# World beliefs, character strengths, and hope for the future

**Robert E. McGrath** [ID] *

School of Psychology and Counseling, Fairleigh Dickinson University, Teaneck, NJ, United States of America

* mcgrath@fdu.edu

## Abstract

Research in recent years has revealed the rate of premature and avoidable deaths from suicide and drug/alcohol misuse is rising in the United States. These are sometimes referred to as deaths of despair based on evidence that they are concentrated in relatively poor communities with less access to social resources and low labor force participation. The pattern was first noted in middle-aged White men but seems to be gradually spreading to other ethnic groups. As a first step in establishing a psychological response to this public health issue, the present article summarizes two studies that compared psychological variables to demographics as predictors of hopefulness. A number of intriguing findings emerged. Despite concerns about American despair and conflict, U.S. residents proved the most hopeful among residents of eight countries. Low-income Americans are particularly hopeful except for low-income Whites. Positive character traits and primal beliefs about the world generally proved to be better predictors of hope than ethnicity, financial status, or their interaction. A number of relationships were found between psychological variables and community demographics. The findings as a group suggest hopefulness is driven more by psychological variables than by life circumstances. It is suggested that psychologists could play an important role in the study of this topic by implementing programs intended to enhance hopefulness in impoverished populations, and by encouraging an intentional communal focus on the importance of enhancing well-being.

## Introduction

Case and Deaton [1] were the first to raise an alarm about the growing rate of preventable deaths among middle-aged Americans who identify themselves as White. The resulting decline in mean life expectancy for the population as a whole is unique to the United States among wealthy countries. They later estimated that an additional 600,000 deaths occurred between 1999 and 2017 as a direct result of suicide or misuse of alcohol or drugs [2]. This pattern has since been extensively studied, with consistent evidence that the increased rate of preventable deaths is particularly characteristic of White men with lower levels of education and financial stability [3–5]. They are particularly prevalent in communities marked by lower levels of well-being and hopefulness [3], with the result that they have come to be labeled "deaths of despair" [2].

(ahinfo@authentichappiness.org). Reasonable requests will be honored."

**Funding:** The author(s) received no specific funding for this work.

**Competing interests:** I have read the journal's policy and the authors of this manuscript have the following competing interests: The author is a Senior Scientist for the VIA Institute on Character, which is the copyright holder for one instrument used in this research. The author does not benefit directly from this instrument.

On the basis of these findings, several researchers have concluded the rising preventable death rate can be traced primarily to a loss of faith in the future due to a decline in opportunities for the less educated, combined with gaps in the country's social safety net [2, 6]. Several studies have expanded on this explanation. Lack of employment may play a particularly central role [4, 7], while Siddiqi et al. [8] suggested the phenomenon may reflect a declining sense of social status among low-income Whites relative to other groups in the population.

It is important to place this issue in the context of pre-existing differences in lifespan across ethnic groups. Mortality rates for Americans who identify themselves as Black and Hispanic remain higher than those for White Americans, though the gap may be narrowing [9]. The recognition of deaths of despair among Whites therefore does not mitigate the importance of remediating existing differences that at least in part reflect disparities in access to healthcare services [10]. That said, a growing sense of dis-ease about the future in a substantial portion of the American population merits attention, especially as recent evidence suggests the rate of preventable deaths is now increasing among non-Whites [11–13].

Psychiatrists, economists, public health experts, and shapers of public policy have all weighed in on this epidemic of despair. In contrast, psychologists have paid little attention to this issue. As of July 2022, PsycINFO offers only 32 hits for the search term "deaths of despair," and almost none of those were published in primarily psychological venues. Our absence from this dialog is unfortunate, since despair and its opposite, hope, at the individual level are very much psychological concepts.

To begin an exploration of how psychological attitudes and traits might play a role in the understanding of American hope and despair, this study was conducted to investigate the relationship between two sets of variables (referred to collectively as the "psychological variables" to simplify the presentation) and hope for the future. These sets are provided in Table 1. The first set consists of what have been labeled world beliefs, or primals [14]. The concept of primals has recently been introduced to provide a framework for evaluating people's characteristic expectations about the world. The primals model includes 22 distinct world expectations called the tertiary primals, such as beliefs about whether the world is changeable or progressing; three specific factors, called the secondary primals; and one general factor called the primary primal.

The second set of psychological variables has been referred to as character strengths. These refer to positive personality traits, elements of character that were identified as having a particularly important role to play in social and personal life in that they contribute to flourishing and well-being within individuals and in others. As a consequence, they tend to be socially admired and valued in people. The most widely referenced model has identified 24 such strengths [15]. Examples from their list include Fairness, Gratitude, and Zest.

These sets seem particularly promising as a starting point for examining potential psychological contributors to a sense of hopefulness about the future, for several reasons. They have both been demonstrated to have important associations with styles of behaving (e.g., [16]), and with positive versus negative emotional states (e.g., [14, 17]). Second, they are considered relatively stable elements of an individual's perceptions of self, the world, and others, and so are likely to play a role in the formation of beliefs about the future. Third, these two set attempt to provide a relatively comprehensive enumeration of the elements involved in positive personality and positive world beliefs.

Previous research has provided some information about relationships between these two sets of psychological variables and demographic variables. Supplementary materials provided by Clifton et al. [14] indicated primal factor scores were relatively unrelated to age, gender (though women had higher scores on Alive than men), and education. Asians tended to demonstrate more negative generate lower scores on all four, and Blacks reported higher scores on

**Table 1. The primals and VIA character strengths.**

| Primals | Character Strengths |
|---|---|
| Good[a] | Appreciation of Beauty and Excellence (Beauty) |
| Safe[b] | Bravery |
| Enticing[b] | Creativity |
| Alive[b] | Curiosity |
| Abundant | Fairness |
| Acceptable | Forgiveness & Mercy (Forgiveness) |
| Beautiful | Gratitude |
| Changing | Honesty |
| Cooperative | Hope |
| Funny | Humility |
| Harmless | Humor |
| Hierarchical | Judgment & Open-Mindedness (Judgment) |
| Improvable | Kindness |
| Intentional | Leadership |
| Interactive | Love of Learning (Learning) |
| Interconnected | Capacity to Love and Be Loved (Love) |
| Interesting | Perseverance |
| Just | Perspective |
| Meaningful | Prudence |
| Needs Me | Spirituality |
| Pleasurable | Self-Regulation |
| Progressing | Social Intelligence |
| Regenerative | Teamwork |
| Stable | Zest |
| Understandable | |
| Worth Exploring | |

Non-superscripted primals are referred to as tertiary primals. Terms in parentheses are used as abbreviations for character strengths in subsequent tables.

[a]General primals factor (primary primal)

[b]Specific primals factor (secondary primals)

Alive. There were also small correlations with personal income. Across all 26 primals, no correlation with a demographic variable exceeded .20.

McGrath et al. [18] looked at significant relationships between demographic variables and VIA character strengths. Women generated significantly higher scores than men on five scales reflecting aesthetic interests and emotional attachments, while men had higher scores on three scales that suggest more social assertiveness. Age and education both correlated with 14 strengths in the range .09 to .25, with older and more educated respondents generally associated with higher scores. The most striking finding was that Blacks generated significantly higher scores than Whites for 16 strengths, and in many cases also generated significantly higher scores than Hispanics or Asians.

Stahlmann and Ruch [19] examined the relationships between primals and character strengths. In general, they found the majority of overlap could be explained by the relationship between the primary primal Good and character strengths, with an average correlation of .30. Only Humility, Judgment, and Prudence correlated poorly with Good.

Less is known about the relationship between the psychological variables and hope. One of the character strength scales is explicitly a measure of the tendency to experience hope, so a

relationship between that scale and specific questions about hope for the future can be assumed. For the remaining scales, there is good evidence that they are predictive of positive affect in general. The Clifton et al. [14] supplementary materials indicate many of the primals correlated substantially with a variety of measures of positive and negative emotional states such as optimism, happiness, and depression. Hausler et al. [20] found the mean correlation between the character strengths and positive emotions was .41, while their mean correlation with positive psychological functioning was .63. However, less is known about the extent to which they are indicative of hope over the general tendency to experience positive emotional states.

Two studies were conducted to explore this issue of the primals and character strengths as predictors of hope, and to further explore their connection to demographic variables that have been highlighted in discussions of deaths of despair. Both studies were deemed not human subjects by the Institutional Review Board of Fairleigh Dickinson University because data were provided to the researcher in an unidentiable form.

The original hypotheses and SPSS syntax for this study are available at https://osf.io/yxjmz. Due to gaps in the data that were unknown prior to conducting this study, some of the hypotheses listed online could not be tested. Only data from Study 1 are available directly from the author. It is noted from the outset that these are purely correlational studies, as is true of the entire empirical literature on deaths of despair to date, but an important issue for the discussion will be the potential for action items and interventions that psychologists can potentially offer to help address this issue.

## Study 1

Study 1 was a direct comparison of demographic variables versus psychological variables as predictors of hopefulness. The main questions that were addressed in this study were the following:

1. Are primal world beliefs and character strengths related to key demographic variables that have been identified as relevant to deaths of despair?

2. Are primals and character strengths predictive of future hopefulness?

3. How do the psychological variables compare with demographics as predictors of future hopefulness?

### Materials and methods

**Participants.** The potential sample consisted of adults who accessed the website of the VIA Institute on Character (https://viacharacter.org) in November 2021-March 2022 for purposes of completing the VIA Inventory of Strengths-Positive (VIA-IS-P) [21], which is a measure of the 24 character strengths. Before doing so, they were shown the site's standard consent form, which indicates their data may be used for research purposes, and they were given access to the site's privacy policy. After agreeing to the consent, they were shown the VIA-IS-P. Upon completing the inventory, they received personalized feedback on their results. A random third of visitors to the site during that period who completed the English language version of the VIA-IS-P were asked if they would be willing to complete an additional set of questions for research purposes. No tally was provided of the number of individuals offered the opportunity to participate, but 11,247 agreed to do so.

The first set of analyses compared results across countries that were represented by at least 200 participants. This reduced the sample size to 8,969. Demographic statistics for this sample

may be found in Table 2. Later analyses focused on residents of the United States ($N = 4,861$). Demographics for this sample are also provided in the table.

**Measures.** The VIA-IS-P is a 96-item questionnaire that consists of 4-item scales for each of the 24 VIA character strengths. Items are completed on a 1–5 scale from *Very Much Unlike*

**Table 2. Demographic statistics.**

| | Study 1 | | | | Study 2 | |
|---|---|---|---|---|---|---|
| | Large-Sample Countries | | U.S. | | | |
| | *N* | *%* | *N* | *%* | *N* | *%* |
| **Age** (*M/SD*) | 34.47 | 13.40 | 33.87 | 13.60 | | |
| 18–20 | | | | | 19843 | 22.12 |
| 21–24 | | | | | 23053 | 25.70 |
| 25–34 | | | | | 23538 | 26.24 |
| 35–44 | | | | | 11834 | 13.19 |
| 45–54 | | | | | 6407 | 7.14 |
| 55–64 | | | | | 3598 | 4.01 |
| > 64 | | | | | 1413 | 1.58 |
| **Location** | | | | | | |
| Australia | 1624 | 18.11 | | | | |
| Canada | 630 | 7.02 | | | | |
| India | 220 | 2.45 | | | | |
| New Zealand | 206 | 2.30 | | | | |
| Philippines | 382 | 4.26 | | | | |
| Singapore | 201 | 2.24 | | | | |
| United Kingdom | 648 | 7.22 | | | | |
| United States | 5058 | 56.39 | | | | |
| **Gender** | | | | | | |
| Men | 2572 | 31.31 | 1497 | 31.94 | 48786 | 69.89 |
| Women | 5643 | 68.69 | 3190 | 68.06 | 21017 | 30.11 |
| **Financial Status** | | | | | | |
| Poor | 113 | 1.32 | 66 | 1.36 | | |
| Struggling | 1056 | 12.32 | 618 | 12.71 | | |
| Comfortable | 6207 | 72.40 | 3467 | 71.32 | | |
| Wealthy | 1197 | 13.96 | 710 | 14.61 | | |
| **Education** | | | | | | |
| < High school | 463 | 5.43 | 181 | 3.73 | 11343 | 11.07 |
| High school | 950 | 11.14 | 451 | 9.29 | 10580 | 10.32 |
| Some college | 1624 | 19.05 | 1143 | 23.55 | 30867 | 30.12 |
| Certificate | 515 | 6.04 | 139 | 2.86 | | |
| Associate's | 446 | 5.23 | 357 | 7.35 | 9791 | 9.55 |
| Bachelor's | 2307 | 27.06 | 1113 | 22.93 | 20679 | 20.18 |
| Graduate work | 2222 | 26.06 | 1470 | 30.28 | 19215 | 18.74 |
| **Ethnicity** | | | | | | |
| American Indian/Alaska Native | | | 23 | 0.48 | | |
| Asian | | | 318 | 6.60 | | |
| Black or African American | | | 448 | 9.30 | | |
| Hispanic or Latino | | | 514 | 10.67 | | |
| Native Hawaiian or Pacific Islander | | | 40 | 0.83 | | |
| Some other race | | | 120 | 2.49 | | |
| White | | | 3355 | 69.63 | | |

*Me* to *Very Much Like Me*. Only total scores were provided for each scale, so reliability statistics could not be calculated for the present sample. However, prior studies have found reliabilities across the 24 scales to be at least adequate (e.g., [22]). The validity of the scales as predictors of scale-relevant behavioral tendencies has also been established [21, 22]. All items on the VIA-IS-P are keyed positively.

The instrument that was originally developed to gauge all 26 primals consists of 99 items. Because participants had just completed the lengthy VIA-IS-P, the website managers asked for a shortened version. Accordingly, an 18-item short form primals measure was administered (PI-18) [23] that only allowed measurement of the four primary and secondary primals. Items are completed on a 0–5 scale from *Strongly Disagree* to *Strongly Agree*. The Good scale (15 items; coefficient $\alpha$ = .85 in the present sample) reflects the general factor that emerged in factor analyses of the original measure. Three more specific factors also emerged indicating beliefs about whether the world is Safe (6 items; $\alpha$ = .80), Enticing (7 items; $\alpha$ = .80), and Alive (5 items; $\alpha$ = .81). Clifton and Yaden [23] provided evidence of test-retest reliability and external validity for the PI-18.

**Procedure.** In addition to the standardized measures and demographic questions, participants completed several individual items from the "Thinking about the Future" survey developed by researchers from The Brookings Institution and Washington University in St. Louis [24]. One item had to do with current financial status. Participants responded on a four-point scale indicating whether they saw themselves as poor, struggling, comfortable, or wealthy. A second was the Cantril [25] ladder item that has been adopted as a standard approach to measuring life satisfaction across countries [26]. Participants were instructed to think of a ladder with 11 steps on which the eleventh or top step represented the best possible life for them, and the bottom step the worst possible life. They then indicated where they thought their life was right now on the ladder.

Two other items had to do with hope for the future. The first was a four-point hope scale on which participants indicated very little hope, a little hope, a fair amount of hope, or a great deal of hope that their future would go well. The second was a Cantril ladder on which participants indicated where on the 11-point scale they thought they would be in approximately five years' time.

The two hope items correlated .51, so they were combined for purposes of generating a more reliable index of hope for the future. Because they were on different scales, the items were standardized and then averaged. The resulting reliability for the two-item scale was $\alpha$ = .68.

Primary analyses focused on the psychological variables, the standardized hope variable, the financial status variable, and Black vs. White ethnicity in the United States sample. Given the substantial sample sizes and resulting power, statistical significance was not considered the most useful indicator of the practical importance of a relationship for these analyses. Significance levels are reported at times. However, variables in primary analyses were treated as dimensional to the extent possible to allow for computation of correlation coefficients. There is substantial evidence available concerning what represents a typical effect in psychological research (e.g., [27, 28]). Semi-partial correlations were used to represent results from multiple regression analysis results.

## Results

**Cross-cultural analyses.** The first analysis compared the eight countries deemed to be sufficiently represented in the sample (see Table 2) on their current life satisfaction and hope for the future. Though this study is primarily about hope in the United States, these results will be used as background for later analyses and discussion of the results.

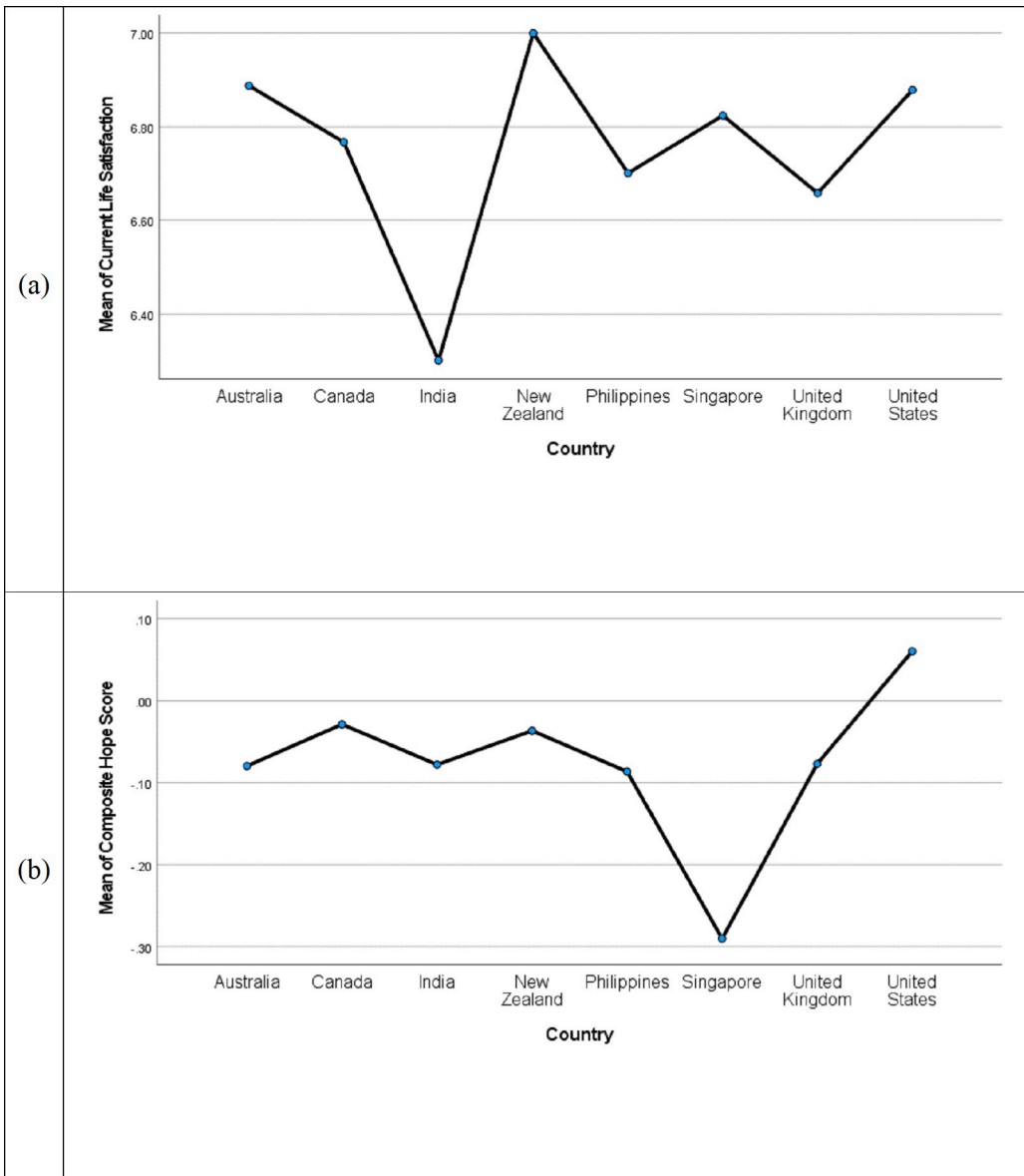

**Fig 1. Life satisfaction and hope scores for eight countries.** (a) Mean current life satisfaction ratings. (b) Mean composite hope for the future ratings.

The omnibus test was significant for the Cantril ladder evaluation of current life, $F(7, 8550)$ = 5.10, $p < .001$ (Fig 1A). Based on least significant difference tests, India's mean life satisfaction was significantly lower than that for any other country in the analysis. Australia, New Zealand, and the United States all reported greater current life satisfaction on average than the United Kingdom. No other differences were significant.

The omnibus test for hope for the future was also significant, $F(7, 8612)$ = 9.80, $p < .001$ (Fig 1B). Interestingly, Americans reported the highest mean level of hope. Based on least significant difference tests, the United States mean was significantly higher than that for any other country except New Zealand.

To explore this relationship further, we examined whether Americans' sense of hope was associated with financial status and education differently than elsewhere. Unfortunately, only

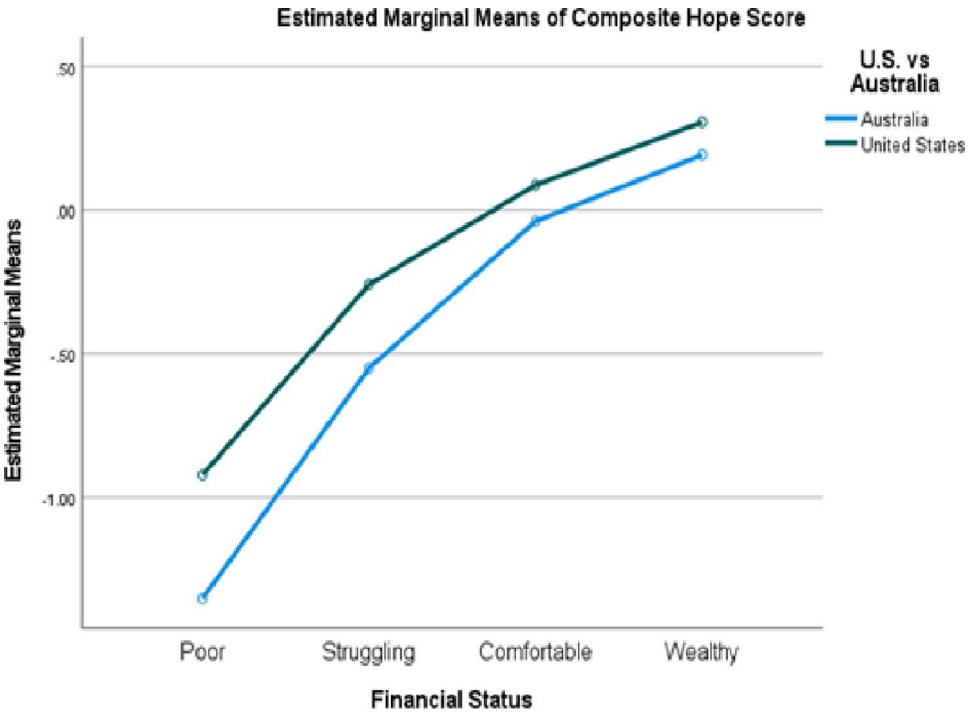

**Fig 2. Hope scores by financial status in the United States and Australia.**

the United States and Australia had enough participants at each level of financial status to be deemed reliable (at least 20 participants per level of financial status). The main effects for financial status and country were significant, but not the interaction (Fig 2). At every level of financial status, the American participants reported greater hope. Post hoc tests revealed Americans who were poor, struggling, or comfortable reported more hope than Australians at the same level; only the wealthy were equally hopeful.

The educational comparison was limited to those countries with at least 20 participants representing each of the bottom two levels of the education variable (high school education or less). This restricted the analysis to Australia, New Zealand, and the United States. Fig 3 combines all groups achieving some educational end point beyond high school to simplify the presentation. Again, both main effects but not the interaction was significant, though there was more variability in the results than for financial status.

**U.S. demographic comparisons.** All subsequent analyses were limited to participants from the U.S. The first analyses focused primarily on bivariate relationships for demographic variables. There was significant variation across the largest ethnic groups (Asian, Black, Hispanic, and White) in terms of education and financial status. Post hoc tests for ethnicity indicated Hispanics were significantly less educated on average than all other groups, and Blacks were less educated than Whites. Blacks reported they were poorer than all other ethnicities, while Hispanics reported they were poorer than Whites and Asians.

Table 3 provides zero-order correlations for demographic variables with the psychological variables and the criterion variables. Based on sources suggesting that correlations between .10 and .25 can be considered medium-sized (e.g., [27]), most relationships were small. Among the character strength variables, the largest correlation for education was that for Love of Learning (.25). Only four others were in the range [.10, .14] (Gratitude, Leadership, Perspective, and Honesty). Gratitude, Hope, and Zest had the strongest relationships with financial

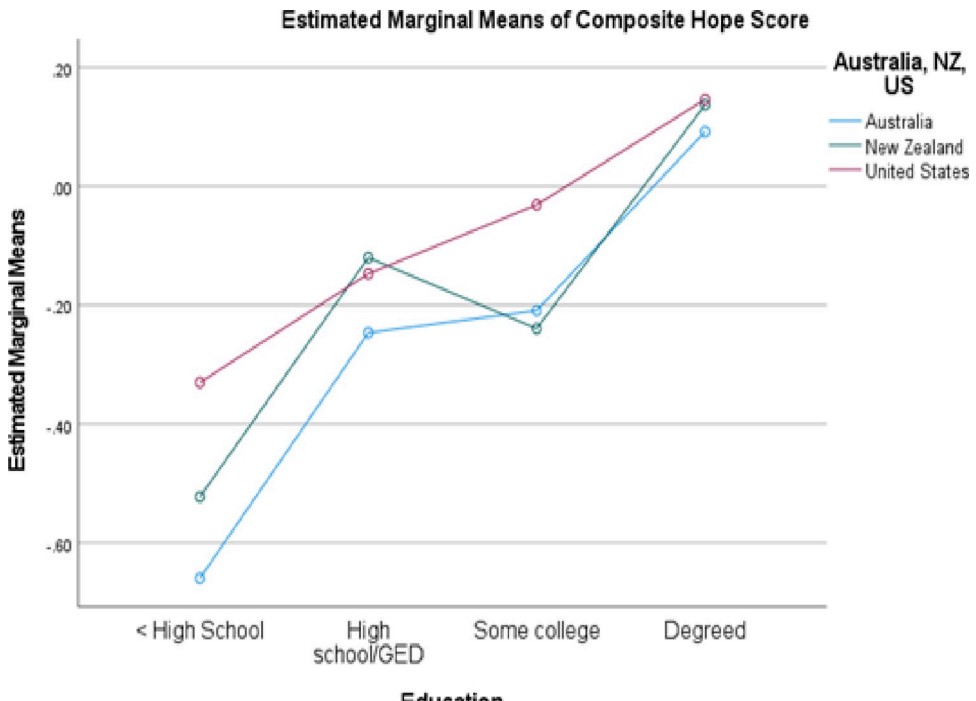

**Fig 3. Comparison of hope scores by education in three countries.**

status. Women reported more Appreciation of Beauty, Love, Kindness, Social Intelligence, and Gratitude than men but less Humor. When ethnicity was reduced to a dichotomous Black-White variable, the standout relationship was that with Spirituality ($r = .20$), the largest in the table. There were five others that fell in the medium-sized range: Creativity, Gratitude, Hope, Humility, and Prudence. In each of these six cases, Blacks scored higher than Whites. In fact, the present findings replicated the general tendency for Blacks to report more character strengths than Whites first noted by McGrath et al. [18], with significant differences on 18 of 24 strengths.

In contrast, the first three demographic variables demonstrated stronger relationships with primary and secondary primal world beliefs. Education and financial status were moderately related to three of four primals (excluding Alive), as was gender (all but Safe). Ethnicity was only related to Alive, a finding that would seem consistent with the difference in Spirituality. Financial status and education were also reasonably related to both life satisfaction and hope for the future. Not unexpectedly, the correlation between these last two variables was .64.

Because financial status was a better predictor of hope than education, the latter will be omitted from subsequent analyses. Fig 4 provides more detailed results on the relationship between finances and ethnicity, this time focusing on the four largest ethnic groups. Ensuring that each mean in the figure reflected at least 30 respondents required combining the poor and struggling groups. Mean hope for Hispanics was higher than that for Whites at all three levels of finances, while Blacks demonstrated higher means except among the wealthy. Interestingly, Asians reported the lowest mean values for hope at all three levels.

There are some interesting comparisons available between Figs 1B and 4. U.S. Blacks who were poor or struggling generated a mean score of -.041 on the composite hope variable, indicating they were more hopeful than the entire sample for any other country except for Canada (-.029) and New Zealand (-.037). In contrast, the mean for poor and struggling Whites (-.374)

**Table 3. Relationships between psychological and key demographic variables.**

|  | Education | Finances | Gender | B-W |
|---|---|---|---|---|
| **Character Strengths** |  |  |  |  |
| Beauty | .04 | -.04 | .17 | .00 |
| Bravery | .01 | -.03 | -.04 | .09 |
| Creativity | .01 | .00 | -.07 | .13 |
| Curiosity | .02 | .06 | -.05 | .06 |
| Fairness | .08 | -.02 | .02 | .03 |
| Forgiveness | .08 | .03 | .01 | .00 |
| Gratitude | .14 | .16 | .12 | .11 |
| Honesty | .12 | .04 | .06 | -.02 |
| Hope | .08 | .10 | .02 | .11 |
| Humility | .05 | -.03 | .00 | .12 |
| Humor | -.04 | .00 | -.11 | .05 |
| Judgment | .07 | .00 | -.09 | .09 |
| Kindness | -.01 | -.01 | .14 | .03 |
| Leadership | .13 | .08 | .00 | .05 |
| Learning | .25 | -.01 | .08 | .05 |
| Love | .05 | .04 | .14 | .03 |
| Perseverance | .07 | .07 | -.03 | .05 |
| Perspective | .13 | .06 | .07 | .09 |
| Prudence | .07 | .02 | .02 | .11 |
| Self-Regulation | .01 | .06 | .00 | .06 |
| Social Intelligence | .00 | .01 | .13 | .06 |
| Spirituality | .07 | -.02 | .09 | .20 |
| Teamwork | .01 | .03 | -.02 | .06 |
| Zest | .03 | .12 | .02 | .08 |
| *M* | .06 | .03 | .03 | .07 |
| **Primals** |  |  |  |  |
| Good | .26 | .16 | .14 | -.03 |
| Safe | .27 | .17 | .09 | -.09 |
| Enticing | .21 | .12 | .11 | -.04 |
| Alive | .02 | .04 | .18 | .12 |
| *M* | .19 | .12 | .13 | .07 |
| **Life Satisfaction** | .20 | .31 | .01 | -.04 |
| **Hope Composite** | .15 | .22 | .07 | .06 |

Correlations with an absolute value > .03 are significant. Gender was coded so that a positive correlation indicates a higher mean for women than men. B-W (Ethnicity) was limited to Blacks and Whites, and coded so that a positive correlation indicates a higher mean for Blacks. Subset means are based on absolute values.

was substantially lower than the mean for any other country; the closest was that for Singapore (-.290). Based on this finding, subsequent demographic analyses will therefore focus on the combination of finances and ethnicity.

**U.S. personality variables and hope.** Table 4 provides results from key analyses evaluating the psychological variables as predictors of the composite hope variable, and comparing it to the key demographic comparison that has been identified in the literature on deaths of despair, Black versus White men who are poor or struggling. The first column presents zero-order correlations. Not surprisingly, the character strength indicating the general tendency to be hopeful was the best single predictor of the composite hope variable. However, the correlations for Gratitude and Zest (which refers to enthusiasm and energy) were also quite high, and

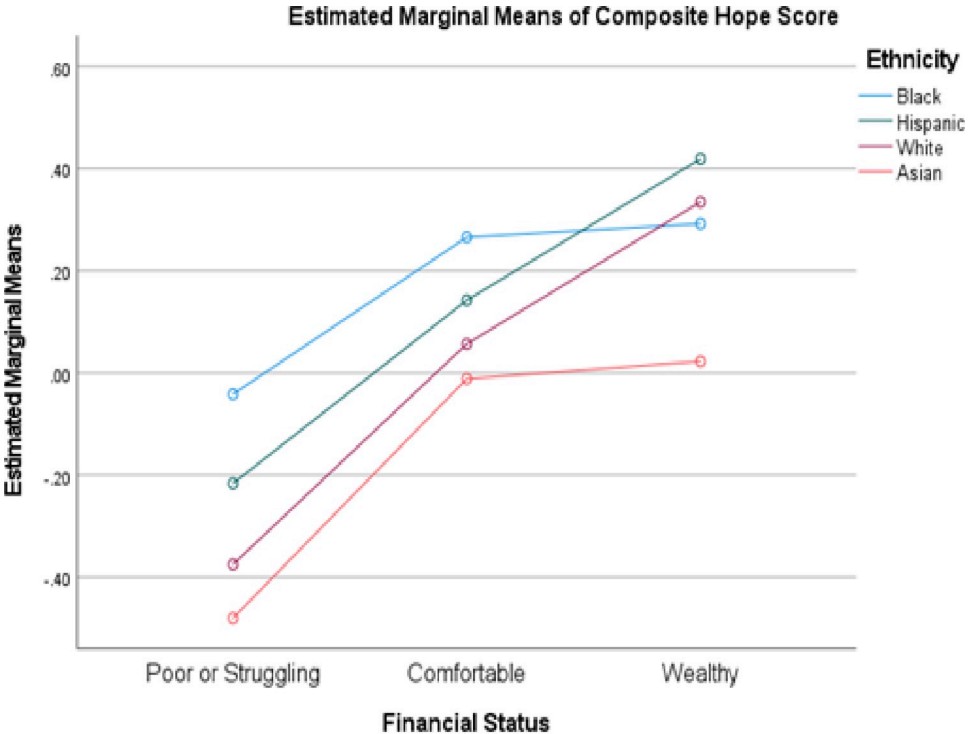

**Fig 4. Comparison of hope scores by ethnicity and financial status in Americans.**

every value in the column was at least medium-sized except that for Humility. Relationships for primals were even larger on average, with a mean correlation of .42, and all four were substantial. In contrast, ethnicity among poor Black and White men was a minor predictor of hope.

The right-hand column in the table was based on multiple regressions that evaluated each of the predictors over and above life satisfaction, to disambiguate hope from the general tendency to see one's life in positive terms. The results are quite consistent across the psychological variables. The mean correlation for both character strengths and primals was cut almost exactly in half. The relationship between psychological state and hope is partly a function of current life satisfaction. Even so, 15 character strengths still correlated .10 or higher with hope, and correlations for the primals remained > .20 except that for Safe.

In contrast, the relationship between financial status and hope disappeared; at .02, it was no longer significant. Financial status only seems to predict hope to the extent that one is presently satisfied with life. One of the most interesting findings was that for poor men, the relationship between ethnicity and hope actually increased, suggesting that current life situation was suppressing of a stronger difference in the tendency to experience hopefulness between Blacks and Whites among poor men.

## Study 2

The second study was conducted to examine relationships between the psychological variables and communities. The inclusion of partial information about location (the first three digits of the U.S. zip code) made it possible to aggregate participants by residential area. This allowed for looking at the psychological variables in the context of key community characteristics. The main questions addressed here were:

**Table 4. Psychological variables as predictors of hope.**

|  | r | sr |
|---|---|---|
| **Character Strengths** |  |  |
| Beauty | .16 | .09 |
| Bravery | .20 | .12 |
| Creativity | .18 | .12 |
| Curiosity | .31 | .16 |
| Fairness | .15 | .05 |
| Forgiveness | .20 | .07 |
| Gratitude | .44 | .18 |
| Honesty | .24 | .10 |
| Hope | .51 | .28 |
| Humility | .07 | .02 |
| Humor | .17 | .10 |
| Judgment | .12 | .06 |
| Kindness | .17 | .10 |
| Leadership | .28 | .15 |
| Learning | .22 | .12 |
| Love | .28 | .14 |
| Perseverance | .25 | .10 |
| Perspective | .27 | .16 |
| Prudence | .12 | .04 |
| Self-Regulation | .24 | .09 |
| Social Intelligence | .22 | .13 |
| Spirituality | .33 | .18 |
| Teamwork | .19 | .07 |
| Zest | .41 | .18 |
| *M* | .24 | .12 |
| **Primals** |  |  |
| Good | .50 | .26 |
| Safe | .36 | .15 |
| Enticing | .43 | .22 |
| Alive | .39 | .23 |
| *M* | .42 | .22 |
| **Finances** | .22 | .02 |
| **Poor Men B-W** | .06 | .14 |

$r$ = zero-order correlation between the variable and hope; $sr$ = semi-partial $r$ value for the psychological variable over and above life satisfaction. Poor Men B-W is Black versus White men who identified themselves as poor or struggling ($N$ = 119). All correlations with an absolute value > .03 were significant.

1. Are primal world beliefs and character strengths related to key demographic variables of communities in which respondents reside?

2. Which demographic variables best account for community differences?

## Materials and methods

**Participants.** The potential sample consisted of adults who accessed the Authentic Happiness website (https://www.authentichappiness.sas.upenn.edu) during an 11-month period to

complete psychological measures in return for personalized feedback. Visitors to the site provide electronic consent for the use of their data in research prior to being administered any questionnaires. All participants in this study had completed either the site's measure of character strengths, the VIA Inventory of Strengths (VIA-IS) [15]; the Primals Inventory-99 (PI-99) [14]; or both. Analyses were conducted separately for the two measures.

Out of 331,061 unique completers of the VIA-IS, there were 102,475 who reported living in the United States and who provided a numeric value in response to a request for the first three digits of their zip code address. There were also 14,299 unique completers of the PI-99, of whom 8,120 indicated living in the United States and provided a valid zip code. There were 4,546 individuals who completed both measures, for a total initial sample of 106,049. Demographics for this group may be found in Table 2.

**Measures.** The VIA-IS was the original inventory developed for the measurement of the 24 character strengths. It consists of 240 items, 10 items per strength. All items are positively keyed. They are completed on the same scale as the VIA-IS-P. Item-level data was available for 97,818 completions. All coefficient alpha values computed from this sample were $\geq .75$.

The PI-99 was similarly the first instrument developed for the measurement of primal world beliefs. It consists of 99 items completed on the same scale as the PI-18. However, the PI-99 provides scales for each of the 26 primals. Based on 8,120 cases for which item data were available, all coefficient alpha values were $\geq .70$.

**Procedure.** Only the first three digits of the zip code was recorded by the site, so respondents were aggregated into three-digit zip code groups. Analyses were based on these aggregates, i.e., each three-digit zip code represented an observation for purposes of analysis. There were 952 unique three-digit zip code groups for those completing the VIA-IS and 769 for those who completed the PI-99. Some zip codes proved to be invalid when compared to Census statistics and were eliminated. To reduce the impact of outliers on the findings, zip code groups with 10 or less VIA-IS completions were omitted from character strength analyses, while at least two completions of the PI-99 were required to be included in the primals analyses. The latter is less stringent than desirable, but more conservative criteria substantially reduced the sample. These two factors limited the final sample to 790 zip code groups for the analyses of the VIA-IS and 643 for the PI-99.

For each three-digit zip code group, Census data for the following variables were downloaded for all corresponding five-digit zip codes: median household income; the percent of families below the poverty line; the number of individuals identifying as either monoracially White, Black, or Asian; and the number identifying as Hispanic. A mean of the median household incomes was computed for each zip code group weighted by the number of households in each zip code. A mean of the percent of families in poverty was computed weighted by the number of families in each zip code. The total number of White, Black, and Asian individuals for the zip code group was divided by the total number of monoracial individuals. The total number of Hispanic individuals for the zip code group was divided by the total number responding to that question.

It should be noted this aggregation of zip codes into zip code groups was mandated by the data, but there was often substantial variability among zip codes within any one group. On average, there are 37.05 five-digit zip codes in the U.S. that begin with the same three digits ($SD = 20.08$). For example, the difference was computed for the minimum and maximum median household incomes across the zip codes included in each zip code group. The maximum difference within a zip code group was $190,133, and the mean of these differences in the medians was $50,428. It was also possible that the data included demographic variables from zip codes not represented by a single person who completed the VIA-IS or the PI-99. The values used here for community demographics therefore should be considered very rough approximations for the individuals who completed either of the psychological measures.

## Results

The left-hand portion of Table 5 provides zero-order correlations between character strengths and community demographics. Despite the limitations of the data, there were a substantial number of correlations that exceeded .10. There was a consistent tendency for lower character strength means in communities with larger proportions of purely White residents. In contrast, the percent of Blacks and the percent of families below the poverty line were associated with a number of moderate correlations in the positive direction, i.e., more Blacks and more poverty in the region were associated with higher scores on character strengths. Most relevant to the current study, the general tendency towards hopefulness was greater in poorer communities and communities with more Blacks, and lower in wealthier communities and those with more Whites. There was some tendency for household income correlations to mirror those for poverty, but the relationships with poverty rate was consistently stronger.

One problem with the interpretation of the results in Table 5 is that the three demographic variables emerging as the best predictors of character strengths—percent monoracially identifying as White, percent Black, and poverty rate—are interrelated variables. To explore the findings further, simultaneous regressions were computed for each character strength using these three demographic variables as predictors. Semi-partial correlations from these analyses can be found in the right-hand side of the table.

**Table 5. Correlations between character strengths and community demographics.**

|  | r | | | | | | sr | | |
|---|---|---|---|---|---|---|---|---|---|
|  | **Household Income** | **White %** | **Black %** | **Asian %** | **Hispanic %** | **Poverty Rate** | **White %** | **Black %** | **Poverty Rate** |
| Beauty | 0.04 | -0.21 | 0.11 | 0.13 | 0.20 | 0.09 | -0.19 | -0.08 | 0.04 |
| Bravery | -0.20 | -0.14 | 0.19 | -0.08 | 0.10 | 0.28 | 0.02 | 0.06 | 0.22 |
| Creativity | 0.00 | -0.23 | 0.24 | 0.03 | 0.15 | 0.17 | -0.07 | 0.07 | 0.08 |
| Curiosity | 0.18 | -0.17 | 0.10 | 0.17 | 0.08 | -0.03 | -0.16 | -0.03 | -0.09 |
| Fairness | -0.15 | -0.12 | 0.17 | -0.04 | 0.08 | 0.23 | 0.01 | 0.06 | 0.17 |
| Forgiveness | -0.04 | -0.09 | 0.09 | 0.00 | 0.02 | 0.05 | -0.02 | 0.04 | 0.01 |
| Gratitude | -0.07 | -0.24 | 0.27 | -0.01 | 0.10 | 0.19 | -0.05 | 0.11 | 0.08 |
| Honesty | -0.23 | -0.08 | 0.17 | -0.11 | -0.01 | 0.26 | 0.09 | 0.11 | 0.21 |
| Hope | -0.11 | -0.29 | 0.27 | 0.01 | 0.16 | 0.25 | -0.11 | 0.05 | 0.14 |
| Humility | -0.20 | -0.13 | 0.17 | -0.10 | 0.09 | 0.24 | 0.01 | 0.06 | 0.19 |
| Humor | -0.02 | -0.09 | 0.16 | -0.05 | 0.00 | 0.11 | 0.06 | 0.13 | 0.06 |
| Judgment | 0.07 | -0.35 | 0.32 | 0.16 | 0.15 | 0.17 | -0.17 | 0.06 | 0.03 |
| Kindness | -0.08 | -0.02 | 0.12 | -0.11 | -0.08 | 0.10 | 0.12 | 0.15 | 0.06 |
| Leadership | -0.07 | -0.20 | 0.23 | 0.01 | 0.11 | 0.21 | -0.03 | 0.08 | 0.13 |
| Learning | 0.03 | -0.20 | 0.16 | 0.07 | 0.10 | 0.10 | -0.11 | 0.01 | 0.03 |
| Love | 0.04 | -0.09 | 0.13 | 0.01 | -0.02 | 0.04 | 0.01 | 0.09 | -0.01 |
| Perseverance | -0.09 | -0.06 | 0.06 | -0.03 | 0.07 | 0.14 | -0.01 | 0.00 | 0.12 |
| Perspective | 0.01 | -0.25 | 0.27 | 0.06 | 0.07 | 0.14 | -0.06 | 0.11 | 0.03 |
| Prudence | 0.00 | -0.26 | 0.24 | 0.11 | 0.12 | 0.18 | -0.11 | 0.04 | 0.08 |
| Self-Regulation | 0.08 | -0.21 | 0.13 | 0.14 | 0.16 | 0.06 | -0.16 | -0.04 | 0.00 |
| Social Intelligence | 0.13 | -0.22 | 0.20 | 0.11 | 0.11 | 0.03 | -0.12 | 0.05 | -0.06 |
| Spirituality | -0.32 | -0.17 | 0.28 | -0.17 | 0.04 | 0.33 | 0.07 | 0.16 | 0.24 |
| Teamwork | -0.07 | -0.13 | 0.11 | 0.02 | 0.10 | 0.18 | -0.06 | -0.01 | 0.14 |
| Zest | 0.06 | -0.22 | 0.20 | 0.07 | 0.14 | 0.09 | -0.10 | 0.04 | 0.01 |
| *M* | -0.04 | -0.17 | 0.18 | 0.02 | 0.09 | 0.15 | -0.05 | 0.06 | 0.08 |

*sr* = semi-partial correlations.

As expected, the effect sizes were substantially reduced, and the results are variable. Looking at the largest of the three semi-partials in the 21 cases where at least one of those correlations was $\geq$ .10 (none of the three predicted Creativity, Forgiveness, or Love well), the numbers are quite similar. The percent of Blacks was the best predictor over the other two in four cases, the percent White in eight cases (all in the negative direction), and poverty rate was the best predictor in nine. Unexpectedly, Spirituality was more related to poverty rate than to percent Black. There are some exceptions, but there is some pattern to which character strengths were predicted best by which demographic variables. Those living in Whiter communities tended to report lower levels of effective personal functioning, e.g., poorer social intelligence, self-regulation, curiosity, and judgment. Participants from communities with a higher proportion of Blacks reported better interpersonal solidarity, e.g., greater gratitude, humor, and kindness. Members of poorer communities reported more resilience and integrity reflected in strengths such as bravery, honesty, leadership, and perseverance.

Table 6 provides the corresponding values for primal world beliefs. There was a tendency for household income to correlate positively and poverty to correlate negatively with world beliefs suggesting safety, but correlations were generally small. There was an interesting tendency for residents of communities with higher proportions of Blacks or Hispanics or more poverty (which are of course correlated) to report more negative world beliefs. Those respondents in more Asian or higher income communities tended to report more positive beliefs.

**Table 6. Correlations between primal world beliefs and community demographics.**

|  | Household Income | White % | Black % | Asian % | Hispanic % | Poverty Rate |
|---|---|---|---|---|---|---|
| Good | 0.12 | 0.01 | -0.06 | 0.10 | -0.01 | -0.09 |
| Safe | 0.19 | 0.00 | -0.05 | 0.13 | -0.01 | -0.14 |
| Enticing | 0.06 | 0.05 | -0.10 | 0.06 | -0.02 | -0.07 |
| Alive | -0.07 | -0.01 | 0.02 | -0.02 | 0.05 | 0.11 |
| Abundant | 0.08 | 0.05 | -0.08 | 0.04 | -0.03 | -0.08 |
| Acceptable | 0.04 | 0.07 | -0.10 | 0.00 | -0.02 | -0.08 |
| Beautiful | 0.02 | 0.12 | -0.12 | 0.00 | -0.08 | -0.10 |
| Changing | 0.00 | 0.01 | -0.01 | 0.00 | -0.02 | 0.01 |
| Cooperative | 0.21 | 0.01 | -0.05 | 0.13 | -0.07 | -0.20 |
| Funny | 0.00 | -0.01 | -0.05 | 0.06 | 0.05 | 0.02 |
| Harmless | 0.20 | 0.05 | -0.11 | 0.10 | -0.04 | -0.19 |
| Hierarchical | -0.16 | -0.02 | 0.06 | -0.11 | 0.07 | 0.16 |
| Improvable | 0.02 | 0.01 | -0.07 | 0.06 | 0.03 | -0.03 |
| Intentional | -0.10 | 0.03 | 0.01 | -0.06 | 0.02 | 0.11 |
| Interactive | -0.05 | 0.02 | -0.04 | -0.03 | 0.04 | 0.08 |
| Interconnected | 0.08 | 0.05 | -0.07 | 0.04 | -0.02 | -0.09 |
| Interesting | 0.12 | 0.03 | -0.08 | 0.10 | -0.04 | -0.11 |
| Just | -0.01 | -0.03 | 0.00 | 0.02 | 0.12 | 0.09 |
| Meaningful | 0.03 | 0.03 | -0.04 | 0.03 | -0.03 | 0.00 |
| Needs Me | 0.02 | -0.05 | 0.03 | 0.05 | 0.02 | 0.03 |
| Pleasurable | 0.15 | 0.03 | -0.08 | 0.11 | 0.00 | -0.12 |
| Progressing | 0.11 | -0.09 | 0.03 | 0.11 | 0.05 | 0.00 |
| Regenerative | 0.06 | 0.01 | -0.01 | 0.04 | -0.02 | -0.02 |
| Stable | 0.18 | -0.04 | -0.01 | 0.12 | -0.02 | -0.15 |
| Understandable | -0.02 | 0.00 | 0.02 | 0.00 | 0.01 | 0.02 |
| Worth Exploring | 0.09 | 0.09 | -0.11 | 0.03 | -0.05 | -0.11 |
| $M$ | 0.05 | 0.02 | -0.04 | 0.04 | 0.00 | -0.04 |

## Discussion

### Summary of the findings

This study offers a number of interesting findings. Though life satisfaction and hopefulness correlate substantially, the first two figures suggest attitudes about the present and the future do not necessarily track each other across countries. The most extreme instances of this conclusion involve India, with the lowest current life satisfaction but typical levels of hope for the future, and more typical levels of current satisfaction but low levels of hope for the future among Singaporeans. Americans as a whole were in the middle of the distribution in terms of current life satisfaction, but they were at the top of the distribution in terms of hope for the future. This finding is particularly surprising given social tensions within American society. For example, Silver et al. [29] reported results from over 16,000 telephone surveys about national in 17 industrialized nations. Among U.S. residents, 90% reported strong political conflict, compared to a median for all 17 countries of 50%. While 71% of Americans saw strong ethnic and racial conflict, the median for all 17 was 48%. Few countries came close, and none exceeded the U.S. sample on either variable. The evidence of continuing hope for the future, particularly among those at low levels of financial stability, is therefore striking. However, the results also suggest poor Whites, and poor White men in particular, do not share this general hopefulness among economically disadvantaged Americans. These findings, combined with evidence that hopefulness among Americans was unrelated to financial status after disentangling the effect of present satisfaction, suggests current life circumstances only partially account for hopefulness.

Psychological variables proved much better at predicting hopefulness for the future. Both primal world beliefs and character strengths predicted hopefulness, but beliefs about the world reflected in the primals played a substantially stronger role in predicting hopefulness than beliefs about self represented by the character strength scales.

The second study suggested community as well as personal demographics could play a role in hopefulness. The correlations between demographics and Hope are some of the largest in Table 5. For example, Hope was one of the strongest correlates with percent White, in the negative direction, and percent Black in the positive direction. This generalization about communities mattering must be proposed tentatively, however. Ethnic, racial, and financial information was not collected from the respondents, so community effects cannot be disentangled from the demographics of the respondents.

An unexpected finding was that the character strengths most strongly correlated with Whiter and Blacker communities were three that are indicative of effective decision-making: Judgment, Perspective, and Prudence. This pattern does not appear in Table 3, suggesting it does have something to do with community. The finding merits further study, as does the unexpected variation in which demographic community variable proved the best predictor of each character strength.

### Implications

As other researchers have noted, low hopefulness by itself is not sufficient to explain the phenomenon of deaths of despair in low-income Whites. The comparison of the United States and Australia in Fig 2 indicates poor ($M$ = -1.352) and struggling (-.550) Australians are experiencing greater levels of despair than Americans at corresponding levels (-.922 and -.259, respectively). Singaporeans as a whole reported greater despair than any other country (-.290). Finally, Asian Americans reported the lowest levels of hopefulness of any U.S. ethnic group. None of these populations has yet been identified as suffering an increased rate of avoidable

deaths. This disparity in rate of despair versus rate of deaths of despair in the United States has to be understood in terms of other contributors to the latter. Previous authors have pointed to greater access to weapons and opioids in the United States than in other countries as an important moderator (e.g., [12, 30]). The finding that 39.0% of American Whites, 18.1% of Blacks, but only 11.8% of those from other ethnicities reported a firearm in their household [31] may represent a protective factor for Asian-Americans and, to a lesser extent, Blacks. Others have suggested the fraying of the American social safety net [2, 32] and the decline in the number of high-prestige blue-collar occupations [4, 7] as major contributors to the relationship between despair and death.

Tables 3 and 5 both provide evidence suggesting that Blacks and members of communities with higher proportions of Blacks consistently report higher levels of character strengths than Whites. This might also provide a protective factor for Blacks from less educated or poorer backgrounds. If so, this could suggest that even if the Black-White disparity in the rate of deaths of despair continues to decline, Blacks may continue to demonstrate greater resilience than Whites. It is an intriguing question whether Blacks' greater sense of character relates to their history of greater disadvantage in the United States, but the present findings unfortunately provide no insight into that question.

This article began by noting the relative absence of discussion of deaths of despair among psychologists. The question that arises at this point is whether psychologists have a unique contribution to make to the dialog. We are not in a position to mitigate loss of employment, repair the social safety network, or reduce access to opioids or firearms among those at risk. However, there are two courses of action worth considering. At the micro level, a variety of interventions have been developed that could be worth implementing and evaluating in communities marked by high levels of despair. For example, a variety of interventions have been developed intended to enhance feelings of gratitude, which was found to be one of the best predictors of hope for the future. These include gratitude journals, letter, or visits [33]. Research on individuals from economically and socially disadvantaged who still report high levels of gratitude or faith in a good world could provide insight into the development of interventions specifically intended for those most at risk for avoidable deaths. Hall et al. [34] evaluated a self-affirmation program specifically developed for poor individuals. They found large effects when compared to controls on cognitive performance tasks. They also found participants in the affirmation treatment pursued information about how to access social benefits at a higher rate. It is important to recognize that most research on the relationship between psychological variables such as the tendency to feel grateful or belief in a good world is based on pre-existing tendencies. We have much less evidence that psychological interventions can reshape these broad elements of our makeup. I am hopeful for what psychological interventions can offer for those who are despairing.

At the macro level, psychologists can be involved in efforts to take on well-being as a national agenda. Bhutan pioneered this concept when, in 1986, its fourth king declared that gross national happiness would be prioritized over gross domestic product in all aspects of government planning. The nation has since implemented an Education for Gross National Happiness Program that will ultimately be incorporated into all schools in the nation. A number of governmental bodies around the world are experimenting with similar efforts, though usually on a more limited scale ([35]). The 2022 World Happiness Report [26] identified three nations that have integrated the goal of advancing well-being into all elements of policy-making: Bhutan, United Kingdom, and New Zealand.

The United States lags behind other countries in the pursuit of well-being enhancement as a societal goal. This seems to reflect American biases. Undoubtedly reinforced by the country's economic successes over the past century, many Americans evince a strong conviction that

private enterprise offers the best means both for ensuring opportunities for personal economic success and for improving social welfare systems such as education and healthcare. One corollary of this belief is that happiness tends to be seen as a purely personal goal rather than one that could be substantially advanced by communal action. That mistrust is unfortunate. Just because the pursuit of happiness is enshrined in the Constitution as an individual right does not mean one has to go it alone.

In summary, the primary takeaways from this study are as follows:

1. Deaths of despair may be considered a largely American phenomenon, but despair may not be. Americans were actually more hopeful than participants from other countries, and this finding was reliable across levels of financial status and education where there were sufficient data. Subsequent points are restricted to Americans.

2. Blacks and Hispanics demonstrated greater hope for the future than Whites and Asian-Americans, who were lowest.

3. Both Black participants and participants living in communities with more Blacks self-report higher levels of character strengths including the tendency to be hopeful. Black participants also reported greater hopefulness for the future.

4. Psychological variables as a group were meaningful predictors of hope for the future, and better predictors than demographic variables that have been associated with deaths of despair. Among character strengths, the best predictors were the general tendency towards hopefulness, zest, and gratitude. The tendency to see the world as good, safe, enticing, and alive were particularly substantial predictors of hope for the future.

5. The first two points suggest that increasing rates of deaths of despair in the United States can only be partly accounted for by the experience of despair. Other factors such as access to opioids and guns as well as general health [32] are also important parts of the puzzle. What this study adds to that discussion is the importance of looking at psychological perspective about self and the world as potential protective factors within those communities susceptible to deaths of despair.

## Limitations

Of course, this study has its limitations, mainly having to do with the quality of the samples. The first is that the method of data collection in both studies limits the generalizability of the sample. In both cases, participants were self-selected accessers of the website who did so for purposes of gaining self-knowledge. An unknown portion of these individuals were udoubtedly referred to the site by therapists, coaches, or significant others. However, the majority are probably self-referred individuals who presumably have a greater than typical interest in obtaining self-knowledge. For example, more than half of the U.S. participants had a college degree or higher, while Census data put this number at 1/3 of the American adult population [36].

There are two sets of statistics that provide some insight into how representative these samples might be. The first are the cross-cultural results for the current-life Cantril ladder. This same question is administered internationally each year as part of the World Happiness Report, which is the most extensive effort available to examine current life satisfaction across nations. The 2023 report provides mean values for the eight countries included in Fig 1 from 2020 to 2022 (see [37], data for their Figure 2.1). Compared to those data (see Table 7 in this article), Singapore did better than expected in the current sample and the United Kingdom

**Table 7. Current results compared to the 2023 World Happiness Report.**

| Country | Current Sample | | World Happiness Report | | |
| --- | --- | --- | --- | --- | --- |
| | *M* | Rank | *M* | Report Rank | 1–8 Rank |
| New Zealand | 7.00 | 1 | 7.12 | 10 | 1 |
| Australia | 6.89 | 2 | 7.10 | 12 | 2 |
| United States | 6.88 | 3 | 6.89 | 15 | 4 |
| Singapore | 6.82 | 4 | 6.59 | 25 | 6 |
| Canada | 6.77 | 5 | 6.96 | 13 | 3 |
| Philippines | 6.70 | 6 | 5.52 | 76 | 7 |
| United Kingdom | 6.66 | 7 | 6.80 | 19 | 5 |
| India | 6.30 | 8 | 4.04 | 126 | 8 |

1–8 Rank refers to the order of means in the World Happiness Report based exclusively on these eight countries.

did worse (though the latter could reflect the relatively recent impact of Brexit). The rank ordering of the other six countries is very consistent across the two data sources, with three occupying exactly the same position in the ordering of the eight countries, and the other discrepancies involving transposition of countries that appear consecutively in the report. Most of the means in the two samples were consistent as well, though the mean ratings for India and the Philippines in the present sample were markedly higher than those in the World Happiness Report.

The results suggest relative placement of means on the current life satisfaction ladder are fairly consistent across the two sources, and in the majority of countries (including the United States) they are also consistent in absolute value. Of course, there is no way of knowing whether the same would be true for ratings of hope for the future, which is the more important variable in the present study. As noted previously, these did not track closely with life satisfaction.

Second, a study was cited earlier that used an online Qualtrics panel sampled to match U.S. adult Census data on the variables gender, age, education, race, and region of the country [18]. Qualtrics was chosen for this task based on evidence that their samples are consistent with those used in the General Social Survey, which is considered the most effective effort at probability sampling in the United States [38]. That study also examined demographic differences in the strengths, and produced results that were very consistent with those of the current study.

Though these offer some hope for the representativeness of the present sample(s), further work with a more neutral and representative sample would be desirable. Another goal for future research might be the collection of a larger sample so that more extensive cross-tabulations could be evaluated, such as adding gender, age, and precise location within the U.S., employment. It would also be interesting to study whether there is more to learn about hopelessness across subgroups of Asian-Americans based on regional heritage.

Second, though the second study raises some intriguing questions about the possible relationship between community demographics and psychological variables, conclusions are compromised by the absence of data on personal demographics. Finally, the degree to which the current samples are reflective of those who are at risk for premature and avoidable death is completely unknown. Prior research on deaths of despair at least has been able to investigate hopelessness in areas where the prevalence of such deaths is relatively high. Future studies might want to include variables that more directly tap into mortality risk in participants such as personal misuse of drugs or suicidal ideation. That said, there is some research suggesting many visitors to the VIA Institute website are at risk for despair, in that prior studies

administering depression questionnaires to research volunteers on the site report that the mean score is indicative of some depression [39, 40]. Though it is a question for which the current data provide no answer, it is possible that individuals seeking self-knowledge are a reasonable source of data for studying some aspects of deaths of despair.

## Acknowledgments

I am grateful to the managers of the VIA Institute on Character and Authentic Happiness websites for their making the datasets used in this study available. I am deeply grateful to Carol Graham for her sage advice throughout this project, and to Jeremy Clifton and Alejandro Adler for their comments on an earlier draft of this article.

## Author Contributions

**Conceptualization:** Robert E. McGrath.

**Formal analysis:** Robert E. McGrath.

**Investigation:** Robert E. McGrath.

**Methodology:** Robert E. McGrath.

**Project administration:** Robert E. McGrath.

**Writing – original draft:** Robert E. McGrath.

**Writing – review & editing:** Robert E. McGrath.

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
