## [Decision Letter · Decision Letter 0]

6 Mar 2023

PONE-D-22-20251World Beliefs, Character Strengths, and Hope for the FuturePLOS ONE

Dear Dr. McGrath

Thank you for submitting your manuscript to PLOS ONE. After careful consideration, we feel that it has merit but does not fully meet PLOS ONE’s publication criteria as it currently stands. Therefore, we invite you to submit a revised version of the manuscript that addresses the points raised during the review process.

We look forward to receiving your revised manuscript.

Kind regards,

Grant Rich, Ph.D.

Academic Editor

PLOS ONE

Journal Requirements:

2. Please ensure that you have specified (1) whether consent was informed and (2) what type you obtained (for instance, written or verbal, and if verbal, how it was documented and witnessed). If your study included minors, state whether you obtained consent from parents or guardians. If the need for consent was waived by the ethics committee, please include this information.

3. Please change "female” or "male" to "woman” or "man" as appropriate, when used as a noun (see for instance https://apastyle.apa.org/style-grammar-guidelines/bias-free-language/gender).

4. Please ensure that you have specified (1) whether consent was informed and (2) what type you obtained (for instance, written or verbal, and if verbal, how it was documented and witnessed). If your study included minors, state whether you obtained consent from parents or guardians. If the need for consent was waived by the ethics committee, please include this information.

Additional Editor Comments (if provided):

Dear Author,

Your paper addresses a topic of considerable importance and utilizes a promising, and intriguing emerging approach regarding the primals, as well as character strengths from the positive psychology paradigm.

A major revision is requested- see the comments from the two reviewers below. In particular I agree with reviewer two that presently the paper leaves the reader struggling to grasp main takeaways and that his reviewer (and I) recommend a revise and resubmit where problems are resolved by streamlining, shortening, and focusing the paper.

Also both reviewers and I agree that more should be said about the sampling- as reviewer 1 notes " Insufficient information is provided to allow a determination of the randomness of the sample vs a convenience sample"

Grant J Rich, PhD

--------------

Reviewer ONE said

This study covers a critical area for US public policy and specifically for behavioral health policy, namely, the tragic increases in deaths of despair in the United States, including suicide and accidental overdose deaths, as well as alcohol involved liver disease. While this has been extensively analyzed by Case and Deaton and the impact of lower levels of education compared to college education documented, the psychological variable of hope as the antithesis of despair has been far less explored despite its clear connection to suicide prevention. For that reason this study makes an important contribution. However, minor revisions are recommended. Insufficient information is provided to allow a determination of the randomness of the sample vs a convenience sample. Specifically, since data was collected by visitors to a website , limited information was provided regarding what brought visitors to the website Second, although at the start of the manuscript the assertion is made that deaths of despair while first documented among whites without a college degree is now being seen among communities of color, this important issue is not meaningfully returned to and related to differential expectations among different races regarding hope for the future and wat has shaped those differential expectations.

Reviewer TWO said

The author explores connections between primals and hope, strengths and hope, and demographics and hope. I read the paper three times and find it a valuable contribution to the nascent primals literature, since many of these relationships have not yet been examined and should be. There are many results here that I can see later papers building on.

However, the paper has two problems. First, I struggled to grasp main takeaways. There are too many exploratory analyses provided that are not united into a clear story. For example, a small set of hypotheses are not specifically stated and returned to in the discussion in a focused way and important results are obscured alongside other results that in my view belong in a supplement (e.g., country level results). The second main difficulty, as the author acknowledges, is that these were not representative samples, large though they might be. Thus, I recommend a revise and resubmit where problems are resolved by streamlining, shortening, and focusing the paper as well as running one additional pre-registered study—a nationally-represented prolific study of ~300 would be sufficient—testing only a handful of the most important hypotheses that the author deems worthy of replication, only measuring those specific primals, strengths, and demographics necessary for doing so, and then discussing only those specific hypotheses in the general discussion, with all other results moved to a supplement. I glanced through the pre-registered hypotheses and feel that a number of them could suffice and the precise direction is a matter of author discretion.

Minor comments/suggestions:

There is at least 1 reference to the “About Me” primal when it is now called “Interactive world belief”, which is referred to correctly in most places I believe.

Stahlmann and Ruch is not in press anymore.

There are a few recent primals papers that the author might not be aware of that may or may not be relevant: https://doi.org/10.1080/17439760.2021.2016907 was the first multi-sample look at primals in connection to life satisfaction and other mental health measures relevant to hope. https://doi.org/10.1111/jopy.12780 shows that the pandemic made a small impact on primals, suggesting connections between demographics and primals might be small. https://doi.org/10.3389/fpsyg.2020.01145 is a theory paper explicitly discussing why relationships between primals and demographics are often small. The relevance of these papers might depend on what sort of streamlined story the author wants to tell.

It may not make sense to report an alpha for a 2 item scale, but that could be a matter of preference. See https://www.researchgate.net/profile/Manfred-Grotenhuis/publication/232610246_The_reliability_of_a_two-item_scale_Pearson_Cronbach_or_Spearman-Brown/links/00b4951759c45468bf000000/The-reliability-of-a-two-item-scale-Pearson-Cronbach-or-Spearman-Brown.pdf.

Per question above on publicly available data: I believe some of the data is not available publicly but this is adequately specified and why.

Sincerely,

Reviewers' comments:

Reviewer's Responses to Questions

**Comments to the Author**

1. Is the manuscript technically sound, and do the data support the conclusions?

Reviewer #1: Yes

Reviewer #2: Yes

2. Has the statistical analysis been performed appropriately and rigorously? 

Reviewer #1: I Don't Know

Reviewer #2: Yes

3. Have the authors made all data underlying the findings in their manuscript fully available?

Reviewer #1: Yes

Reviewer #2: Yes

4. Is the manuscript presented in an intelligible fashion and written in standard English?

Reviewer #1: No

Reviewer #2: Yes

5. Review Comments to the Author

Reviewer #1: This study covers a critical area for US public policy and specifically for behavioral health policy, namely, the tragic increases in deaths of despair in the United States, including suicide and accidental overdose deaths, as well as alcohol involved liver disease. While this has been extensively analyzed by Case and Deaton and the impact of lower levels of education compared to college education documented, the psychological variable of hope as the antithesis of despair has been far less explored despite its clear connection to suicide prevention. For that reason this study makes an important contribution. However, minor revisions are recommended. Insufficient information is provided to allow a determination of the randomness of the sample vs a convenience sample. Specifically, since data was collected by visitors to a website , limited information was provided regarding what brought visitors to the website Second, although at the start of the manuscript the assertion is made that deaths of despair while first documented among whites without a college degree is now being seen among communities of color, this important issue is not meaningfully returned to and related to differential expectations among different races regarding hope for the future and wat has shaped those differential expectations.

Reviewer #2: The author explores connections between primals and hope, strengths and hope, and demographics and hope. I read the paper three times and find it a valuable contribution to the nascent primals literature, since many of these relationships have not yet been examined and should be. There are many results here that I can see later papers building on.

However, the paper has two problems. First, I struggled to grasp main takeaways. There are too many exploratory analyses provided that are not united into a clear story. For example, a small set of hypotheses are not specifically stated and returned to in the discussion in a focused way and important results are obscured alongside other results that in my view belong in a supplement (e.g., country level results). The second main difficulty, as the author acknowledges, is that these were not representative samples, large though they might be. Thus, I recommend a revise and resubmit where problems are resolved by streamlining, shortening, and focusing the paper as well as running one additional pre-registered study—a nationally-represented prolific study of ~300 would be sufficient—testing only a handful of the most important hypotheses that the author deems worthy of replication, only measuring those specific primals, strengths, and demographics necessary for doing so, and then discussing only those specific hypotheses in the general discussion, with all other results moved to a supplement. I glanced through the pre-registered hypotheses and feel that a number of them could suffice and the precise direction is a matter of author discretion.

Minor comments/suggestions:

There is at least 1 reference to the “About Me” primal when it is now called “Interactive world belief”, which is referred to correctly in most places I believe.

Stahlmann and Ruch is not in press anymore.

There are a few recent primals papers that the author might not be aware of that may or may not be relevant: https://doi.org/10.1080/17439760.2021.2016907 was the first multi-sample look at primals in connection to life satisfaction and other mental health measures relevant to hope. https://doi.org/10.1111/jopy.12780 shows that the pandemic made a small impact on primals, suggesting connections between demographics and primals might be small. https://doi.org/10.3389/fpsyg.2020.01145 is a theory paper explicitly discussing why relationships between primals and demographics are often small. The relevance of these papers might depend on what sort of streamlined story the author wants to tell.

It may not make sense to report an alpha for a 2 item scale, but that could be a matter of preference. See https://www.researchgate.net/profile/Manfred-Grotenhuis/publication/232610246_The_reliability_of_a_two-item_scale_Pearson_Cronbach_or_Spearman-Brown/links/00b4951759c45468bf000000/The-reliability-of-a-two-item-scale-Pearson-Cronbach-or-Spearman-Brown.pdf.

Per question above on publicly available data: I believe some of the data is not available publicly but this is adequately specified and why.

Sincerely,

Jer Clifton

6. PLOS authors have the option to publish the peer review history of their article (what does this mean?). If published, this will include your full peer review and any attached files.

Reviewer #1: No

Reviewer #2: **Yes: **Jer Clifton

====

Grant J. Rich, PhD LMT BCTMBPresident-Elect Society for Peace, Conflict, and Violence (APA D48)President-Elect Society for Media Psychology and Technology (APA D46)

Fellow, Association for Psychological Science (APS)Fellow, American Psychological Association (APA)Senior Contributing Faculty, Walden UniversityDr. Rich's SPN Website: http://rich.socialpsychology.org/**Book Website** (Rich, Gielen, & Takooshian, 2017)http://www.infoagepub.com/products/Internationalizing-the-Teaching-of-Psychology**Book Website** (Rich & Sirikantraporn, 2018)https://rowman.com/ISBN/9781498554831/Human-Strengths-and-Resilience-Cross-Cultural-and-International-Perspectives#**Book Website** (Rich, Jaafar, & Barron, 2020) Psychology in Southeast Asia. Routledge.https://www.routledge.com/Psychology-in-Southeast-Asia-Sociocultural-Clinical-and-Health-Perspectives/Rich-Jaafar-Barron/p/book/9780367492144**Book Website** (Rich & Ramkumar, 2022) Psychology in Oceania and the Caribbean, Springerhttps://link.springer.com/book/10.1007/978-3-030-87763-7#editorsandaffiliations **Book Website**(Rich, Kuriansky, Gielen, & Kaplan, in press) * Psychosocial Experiences and Adjustment of Migrants: Coming to the USA, Elsevier**https://www.elsevier.com/books/psychosocial-experiences-and-adjustment-of-migrants/rich/978-0-12-823794-6* **Book **(Rich, Kumar, & Farley, in contract)* Handbook of **Media Psychology and Technology-The Science and the Practice,** Springer*

**============**

---

## [Editor Report · Decision Letter 1]

18 May 2023

World Beliefs, Character Strengths, and Hope for the Future

PONE-D-22-20251R1

Dear Dr. McGrath

We’re pleased to inform you that your manuscript has been judged scientifically suitable for publication and will be formally accepted for publication once it meets all outstanding technical requirements.

Kind regards,

Grant Rich, Ph.D.

Academic Editor

PLOS ONE

Additional Editor Comments (optional):

In my view the author has sufficiently responded to reviewers' feedback and thus the paper is accepted.

Reviewers' comments:

Grant J. Rich, PhD LMT BCTMB LSW 

President-Elect Society for Peace, Conflict, and Violence (APA D48)

President-Elect Society for Media Psychology and Technology (APA D46) Fellow, Association for Psychological Science (APS)

Fellow, American Psychological Association (APA)

Senior Contributing Faculty, Walden University

Dr. Rich's SPN Website: http://rich.socialpsychology.org/

Book Website (Rich, Gielen, & Takooshian, 2017)

http://www.infoagepub.com/products/Internationalizing-the-Teaching-of-Psychology

Book Website (Rich & Sirikantraporn, 2018)

https://rowman.com/ISBN/9781498554831/Human-Strengths-and-Resilience-Cross-Cultural-and-International-Perspectives#

Book Website (Rich, Jaafar, & Barron, 2020) Psychology in Southeast Asia. Routledge.

https://www.routledge.com/Psychology-in-Southeast-Asia-Sociocultural-Clinical-and-Health-Perspectives/Rich-Jaafar-Barron/p/book/9780367492144

Book Website (Rich & Ramkumar, 2022) Psychology in Oceania and the Caribbean, Springer

https://link.springer.com/book/10.1007/978-3-030-87763-7#editorsandaffiliations

Book Website(Rich, Kuriansky, Gielen, & Kaplan, in press)  Psychosocial Experiences and Adjustment of Migrants: Coming to the USA, Elsevier

https://www.elsevier.com/books/psychosocial-experiences-and-adjustment-of-migrants/rich/978-0-12-823794-6

Book 

(Rich, Kumar, & Farley, in contract)* Handbook of*
*Media Psychology* and Technology-The Science and the Practice,* Springer*

---

## [Editor Report · Acceptance letter]

21 Jun 2023

PONE-D-22-20251R1 

World beliefs, character strengths, and hope for the future 

Dear Dr. McGrath:

I'm pleased to inform you that your manuscript has been deemed suitable for publication in PLOS ONE. Congratulations! Your manuscript is now with our production department. 

Kind regards, 

on behalf of

Dr. Grant Rich 

Academic Editor

PLOS ONE